# On the Importance of Acidity in Cancer Cells and Therapy

**DOI:** 10.3390/biology13040225

**Published:** 2024-03-29

**Authors:** Alaa Tafech, Angélique Stéphanou

**Affiliations:** Université Grenoble Alpes, CNRS, UMR 5525, VetAgro Sup, Grenoble INP, TIMC, 38000 Grenoble, France

**Keywords:** acidosis, pH regulation, resistance to acidity, temozolomide, Warburg effect

## Abstract

**Simple Summary:**

Cancer cells often show high levels of use of glucose, leading to acidic conditions around tumors. This acidity can make cancer more aggressive, spreading to other parts of the body and worsening patients’ outcomes. The imbalance in acidity can be both a cause and a result of changes in some cell functions. This review looks into why cancer cells become acidic and how they adapt to survive in this harsh environment. It also discusses the challenges of measuring acidity inside cells, which is crucial to understanding how cells regulate acidity. Finally, it shows how acidity affects tumor growth and whether it helps or hinders cancer treatments.

**Abstract:**

Cancer cells are associated with high glycolytic activity, which results in acidification of the tumor microenvironment. The occurrence of this stressful condition fosters tumor aggressiveness, with the outcome of invasiveness and metastasis that are linked to a poor clinical prognosis. Acidosis can be both the cause or consequence of alterations in the functions and expressions of transporters involved in intracellular acidity regulation. This review aims to explore the origin of acidity in cancer cells and the various mechanisms existing in tumors to resist, survive, or thrive in the acidic environment. It highlights the difficulties in measuring the intracellular pH evolution that impedes our understanding of the many regulatory and feedback mechanisms. It finally presents the consequences of acidity on tumor development as well as the friend or foe role of acidity in therapy.

## 1. Origin of Tumor Acidity

A striking but common feature of the tumor microenvironment is a high level of acidity [1,2,3]. Tumor acidity primarily originates from the increased glycolytic activity of cancer cells. This heightened metabolic process leads to the accumulation of lactic acid in the tumor microenvironment, resulting in acidification, but is glycolysis—or, more exactly, anaerobic glycolysis—the only driver of the increased acidity?

Glycolysis is indeed the common starting point for the two main pathways of energy metabolism: mitochondrial respiration and fermentation, which depend on the presence and absence of oxygen, respectively (Figure 1). In the presence of oxygen, most differentiated cells metabolize glucose to pyruvate by glycolysis, then completely oxidize most of this pyruvate to carbon dioxide during the Krebs cycle in the mitochondrial compartment. It is the mitochondrial respiration or oxidative phosphorylation (OXPHOS), which produces the largest amount of ATP by metabolizing glucose. Conversely, anaerobic glycolysis, or fermentation, is the reaction that converts glucose to lactic acid in the absence of oxygen, producing large amounts of lactate and little ATP (Figure 1). Although glycolysis produces less ATP per mole of glucose than OXPHOS, it allows faster generation of ATP [4].

Cells in the proliferative state need energy and organic resources in order to reproduce the components of the cell (DNA, membranes, and proteins). Glucose is essential because, apart from ATP, it also provides the metabolic intermediates that take part in anabolic reactions. In the case of tumor cells, the majority develops a more glycolytic than oxidative metabolism, often accompanying a greater aggressiveness of these tumors. This led Warburg to suspect a deficit in mitochondrial function responsible for this metabolic orientation [5,6]. It now appears that mitochondria function normally in tumor cells and that the blocking of OXPHOS constitutes an adaptive event (Figure 1).

**Figure 1 biology-13-00225-f001:**
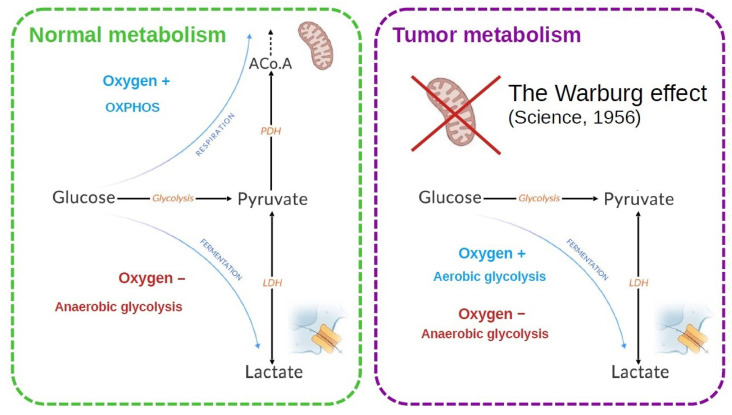
Energy metabolism is classically described for a normal and a tumor cell. In the normal cell, the two metabolic modes depend on the presence of oxygen. In tumor cells, an increased lactate production is observed. Warburg [6] made the hypothesis that the mitochondria were defective to explain this observation. It was later shown that this is not necessarily the case.

In addition to the role of glucose, oxygen is an essential molecule for cell survival since it is used as a final electron acceptor during the production of energy by mitochondrial respiration. Oxygen can diffuse up to a distance of about 100 μm from a blood vessel, and this is why the presence of many vessels is necessary to supply all the cells of a tissue [7,8]. As it develops, the tumor destroys the microvascular network which ensures the physiological and homogeneous oxygenation of the tissue. Tumor cells that divide very quickly are therefore large consumers of oxygen and nutrients and will then quickly run out of resources. They then enter a quiescent state, where they drastically reduce their metabolism and no longer divide. The activation of the Hypoxia-Inducible Factor (HIF) family of transcription factors represents one of the main oxygen-responsive signaling pathways that allow the adaptation of tumor cells to this hypoxic environment [7] (Figure 2). Therefore, hypoxic tumor cells activate the HIF pathway that leads to the production of VEGF (vascular endothelial growth factor), which stimulates the growth of new vessels. The tumor is then vascularized by the process of angiogenesis or neovascularization. Some cells return to the proliferative state where they can divide, and the tumor can thus continue to grow. While hypoxia is harmful to both healthy and tumor cells—where healthy cells can cope with hypoxia through anaerobic glycolysis for a limited time—tumor cells have the ability to adapt to it in several ways and even seem to take advantage of the tumor microenvironment to promote their growth and dissemination via various interactions [9].

Moreover, when cells experience low oxygen levels (hypoxia), their signaling pathways, particularly HIF signaling, shift energy production from oxygen-dependent respiration to fermentation. This allows cells to keep generating ATP despite the lack of oxygen. In hypoxic tumors, this metabolic shift results in increased glycolytic activity, leading to the secretion of large amounts of lactate and acidic hydrogen ions (H^+^). This secretion alters the intracellular pH (pH_*i*_) of cancer cells. To cope with the excess lactate and H^+^ ions, tumor cells activate various proteins involved in regulating the pH, such as CAIX, NHE1, MCT4, and V-ATPase. These proteins help export lactate and H^+^ ions out of the cell, maintaining the pH_*i*_ within normal levels. However, once outside the cell, lactate and H^+^ ions contribute to a decrease in extracellular pH (pH_*e*_, resulting in acidification of the tumor microenvironment.

The combination of hypoxia and acidity creates a chronic hostile microenvironment that directly participates in the adaptation of the tumor cells through phenotypic plasticity [10] and acquired genetic instability. Whereas the reaction to hypoxia proceeds similarly to most cell types whether normal or tumoral, the building up of acidity is enhanced by self-reinforcing mechanisms that kill normal cells and select for and promote tumor aggressiveness. For example, the acidity-induced activation of the NHE1 protein leads to the alkalinization of the intracellular pH, which will participate in the activation of glycolytic enzymes, inhibition of OXPHOS through impaired mitochondria functioning, and cell proliferation [11] (Figure 2). Collectively, these events will increase aerobic glycolysis and environmental acidity, which, in turn, will promote NHE1 overexpression and sustain the cycle of events thus worsening the conditions. The onset of angiogenesis temporarily reduces hypoxia, which leads to HIF oscillations associated with an increase in genetic instability [12].

## 2. Regulation of Intracellular Acidity

It is often reported in the literature that tumor cells are characterized by an inverse pH gradient compared to normal cells, but what does this really mean? The tumor microenvironment is characterized by a high level of acidity. As a consequence, the tumor cells, to survive, activate some regulation mechanisms to maintain an intracellular pH close to the physiological value of about 7.4 in order to keep functioning [13,14,15]. Typically alkaline pH_*i*_ values from 7.1 to 7.6 are measured for acidic pH_*e*_ values from 6.2 to 6.9 [16]. On the other hand, normal cells experiment with an extracellular physiological pH_*e*_, and their pH_*i*_ is usually found to be lower, around 7.0–7.2 [17,18]. Therefore, the reverse pH gradient is not a property of the tumor cells, as we might be led to think, but rather it is merely an observation. Normal cells are rarely observed in acidic conditions, but, if artificially submitted to acidic conditions, they also activate the same regulatory mechanisms. Reciprocally, tumor cells immersed in alkaline pH are also very efficient in restoring an intracellular physiological pH [19].

The reverse pH for tumor cells is, therefore, a misleading concept, since any cell—tumoral or normal—in acidic conditions will try to maintain an intracellular physiological pH, with a pH_*i*_ higher than the pH_*e*_. Reciprocally, in alkaline conditions (above 7.0), the cell will also try to keep its internal physiological pH, this time with a reverse situation in which the pH_*i*_ tends to be lower than the pH_*e*_. So, the reverse pH gradient is not a property of the tumor cell per se but is entirely contextual.

As a consequence, another question is: do tumor cells regulate intracellular acidity in a different way than normal cells? Moreover, do they regulate it more efficiently? To answer these questions, we will first recall the mechanisms involved in this regulation and then discuss the tumor *versus* normal cells’ differences.

The pH regulating proteins comprise Na^+^/H^+^ exchangers (NHEs) [20], V-ATPase [21], the monocarboxylate transporters (MCTs) [22], Carbonic Anhydrases (CA) [20,23], and the ${\mathrm{HCO}}_{3}^{-}$ transporters [24] (Figure 3). The role of each regulator is detailed in the following paragraphs.

### 2.1. The Family of Na^+^/H^+^ Exchangers (NHEs)

Sodium-proton exchangers play a major role in regulation of the pH_*i*_. These Na^+^/H^+^ exchangers allow the electroneutral exchange of a Na^+^ for an H^+^. A 2:2 stoichiometry has also been proposed [25]. This family includes nine isoforms (NHE1 to NHE9) with varied subcellular and tissue distributions. Only three have been identified in the regulation of pH_*i*_: NHE1, NHE2, and NHE3. NHE2 and NHE3 have been localized only in the intestine and kidney, while NHE-1 is abundantly expressed throughout the body. NHE1 is localized at the level of the plasma membrane of the cell, and its role is to expel an H^+^ ion in exchange for the entry of a sodium Na^+^ ion, thereby increasing the pH_*i*_. NHE1 has a great potential to maintain a normal pH_*i*_. Chiche et al. (2010) [3] reported the following experiment: the injection of a large number of H^+^ ions into hamster cells expressing NHE1, NHE2, and NHE3 gives rapid recovery to a normal pH_*i*_, suitable for their survival. Contrariwise, mutated cells in which a lack of NHE1 is observed did not recover their pH_*i*_ after the injection of H^+^ and died in less than an hour due to the persistent acidification. The ubiquitous NHE1 isoform is the only one whose activity has been associated with several properties of cancer cells [26,27,28].

### 2.2. The Enzyme V-ATPase

The vacuolar-type proton pumping ATPase (V-ATPase) is a multiprotein complex using the energy of ATP hydrolysis to electrogenically transport protons [29,30]. It is an enzyme catalyzing the following reaction:

ATP + H_2_O + H ^+^_*intracellular*_ → ADP + phosphate + H ^+^_*extracellular*_

V-ATPase is expressed at the plasma membrane of cancer cells where it has a role in regulating the pH_*i*_ [31,32] and, consequently, contributes to modification of the pH_*e*_. Its expression at the plasma membrane level and its activity are found in highly invasive cancer cells, and it is proposed that its activity is correlated with the aggressiveness of cancer cells [33,34].

### 2.3. Proton-Lactate Transporters (MCTs)

Monocarboxylate transporters (MCTs) allow the transport of lactate, ketone bodies, or pyruvate. This family includes 14 isoforms, of which only the first four play a role in pH_*i*_ regulation. The MCT1, MCT2, MCT3, and MCT4 isoforms are electroneutral membrane cotransporters of protons and lactate with a stoichiometry of 1:1. In cancer cells, significant expression of MCT1 and MCT4 has been demonstrated. MCT4 allows the export of lactate produced by glycolytic (hypoxic) cancer cells, which will then be captured by MCT1 expressed in oxidative cancer cells located at the periphery of the tumor [35]. MCT1 allows the export of lactate outside the cell and also the import inside it. In contrast, MCT4 allows the export of lactate outside the cell but is unsuitable for importing it. MCT1 and MCT4 have a role in the invasiveness of lung cancer cells [36], and MCT4 is involved in the migration of breast cancer cells [37]. MCT1 expression has been shown to be correlated with glioma progression [38].

### 2.4. Carbonic Anhydrases (CA)

The regulation of cell pH involves, in addition to protons, bicarbonate ions. The carbon dioxide/bicarbonate buffer system helps regulate pH and is the most important physiological buffer system. Carbonic anhydrases are able to catalyze the reaction of CO_2_ with H_2_O to produce bicarbonates and protons: CO_2_ + H_2_O ⇌ H^+^ + ${\mathrm{HCO}}_{3}^{-}$. These bicarbonates can be transported into cells to neutralize protons and increase intracellular pH. The transport of bicarbonate ions can be carried out by various co-transporters or exchangers using different associated ions.

Carbonic anhydrases form a superfamily comprising 16 isoforms that catalyze the reversible hydration of CO_2_ into ${\mathrm{HCO}}_{3}^{-}$ and H^+^ and are mainly involved in gas exchange. They can be anchored in the plasma, cytoplasmic, or mitochondrial membranes, or they can even be secreted [39,40]. Among the active enzymes, five carbonic anhydrases are cytosolic (CAI, CAII, CAIII, CAVII, and CAXIII), five are membrane-associated and oriented toward the extracellular environment (CAIV, CAIX, CAXII, CAXIV, and CAXV), two are mitochondrial (CAVA and CAVB), and CAVI is secreted. Three are devoid of catalytic activity (CAVIII, CAX, and CAXI). CAs are involved in many physiological phenomena including respiration and transport of CO_2_ and ${\mathrm{HCO}}_{3}^{-}$ to the tissues and the lungs, pH and CO_2_ homeostasis, certain biosynthetic reactions, and bone formation or resorption.

High expression of CAIX and CAXII has been found in breast cancer cells [41,42] and in many other types of cancers [3]. CAIX and CAXII are membrane-bound, and their catalytic site is oriented toward the outside of the cell. Thus, CO_2_, produced by cellular respiration and which diffuses freely outside the cell, is hydrated into protons and bicarbonate ions (${\mathrm{HCO}}_{3}^{-}$). This action participates in the acidification of the extracellular environment [43] but has also been shown to regulate pH_*i*_ [44,45,46,47]. The expression of CAIX and CAXII is induced by hypoxia via the HIF-1 transcription factor [44]. Their expression can also be increased even in normoxia in the case of mutations making HIF-1*α* constitutively active [48,49]. In addition, the activity of CAIX is increased by an acid medium such as that found in tumors [50]. In hypoxia, induction of CAIX expression contributes to acidifying the extracellular environment of cancer cells and therefore participates in pH regulation [44,51].

An intracellular carbonic anhydrase, CAII, also seems to be found in certain cancers such as brain cancers [52] and to be associated with a poor prognosis in medulloblastomas [53], while its expression is associated with better survival in gastrointestinal stromal tumors [54]. There are also compensation relationships since the reduction in CAIX expression induces an increase in CAII expression and vice versa [55]. Particular attention should be given to CAIX since this isoform is found very infrequently in healthy tissues (in certain parts of the gastrointestinal tract) but is expressed in many tumor tissues, probably due to its hypoxic induction as well as the selective advantage it confers on tumor cells that express it [56].

### 2.5. ${\mathrm{HCO}}_{3}^{-}$ Transporters

These proteins fall into three main categories: electroneutral ${\mathrm{Cl}}^{-}$/${\mathrm{HCO}}_{3}^{-}$ exchangers belonging to the SLC4 family, Na^+^/${\mathrm{HCO}}_{3}^{-}$ co-transporters from the NBC family, and anion transporters within the SLC26 family. These transporters allow the importation of ${\mathrm{HCO}}_{3}^{-}$ bicarbonates which are the main buffer species for H^+^ hydrogen ions, thus contributing to alkalizing the cytoplasm of the cell.

Little is known about the involvement of bicarbonate transporters in cancer cells. These transporters have been proposed to be involved in the pH regulation of cancer cells in different reviews [24,57,58]. Their involvement was initially studied by the use of a pharmacological inhibitor, DIDS, which is not very specific for a single transporter and also inhibits channels permeable to ${\mathrm{Cl}}^{-}$ ions. This inhibitor decreases the pH_*i*_ of cancer cells in vitro [59] and decreases tumor growth in vivo but has a toxicity limiting its potential therapeutic use [60].

Different cellular pathways can be established in cells exhibiting varying degrees of invasiveness or metastatic abilities [34,61]. Moreover, research has shown variations in the subcellular distribution of pH regulators within brain tumor models [62].

From this brief survey on the pH-regulating proteins, it is evident that they consistently exhibit elevated expression levels in tumor cells compared to normal cells. This heightened expression can be attributed to a variety of interrelated factors, including the hostile hypoxic and acidic tumor microenvironment, heightened metabolic demands, dysregulated signaling pathways, and increased genetic alterations.

## 3. Consequences of Acidity

Acidic conditions within the tumor microenvironment promote tumor progression by enhancing cell proliferation, migration, and many other events that contribute to facilitating the selection of more aggressive and invasive cancer cell phenotypes.

### 3.1. Acidity and Tumor Invasion

It has been established that extracellular acidity plays an important role in the process of tumor invasion. Acidity promotes tumor invasion by two main mechanisms. On the one hand, acidity induces toxicity on normal cells, whereas tumor cells are resistant to acid stress; as normal cells die, tumor cells continue to proliferate and invade the open space. On the other hand, the acidity induces the degradation of the extracellular matrix via the release and activation of proteases, such as cathepsin B [63,64], and of matrix metalloproteases (MMPs), such as MMP-1, 2, and 9 [65,66], which are generally believed to be involved in local invasion and tissue remodeling. In a xenograft model with breast and colon cancer cells, Estrella et al. (2013) [67] observed a correlation between acidity and tumor invasion. In tumors, areas with the highest invasion corresponded to areas with the lowest pHs. Interestingly, systematic treatment with sodium bicarbonate decreased the proton gradient in the tumor and prevented tumor invasion [68].

### 3.2. Acidity and Cellular Plasticity

Although the role of extracellular acidity in regulating the stem state of cancer cells has been less characterized, there are a few studies on the regulation of cellular plasticity by acidity. The acidic condition was shown to stimulate the development of a cancer stem cell phenotype [69]. Exposure of glioma cells to an acidic culture medium (pH 6.5) promotes the expression of stem cell markers, such as Olig2, Oct4, and Nanog, and increases the secretion of angiogenic factors, such as VEGF. This happens, in particular, through increasing the expression of HIF-2*α*, which is involved in the phenotype of stem cells and the growth of tumor cells. Additionally, lactate has been shown to attract human mesenchymal stem cells to tumor cells and heighten stem cell migration [70].

### 3.3. Acidity and Immunosuppression

During their development, malignant tumors are able to evade the response of the immune system [71]. Recent studies have shown that the anti-tumor immune response is modulated by the tumor microenvironment, including acidity. Lactic acid produced by tumor cells inhibits the production of cytokines from cytotoxic T lymphocytes in vitro [72]. Activated T cells themselves use glycolysis and rely on the efficient secretion of lactic acid. Lactic acid blocks the efflux of lactate produced in the tumor environment and thus disrupts T cell metabolism. In another study, treatment with the proton pump inhibitor esomeprazole, by reducing tumor acidity, restored the functioning of T lymphocytes [73].

### 3.4. Acidity and Warburg Effect

The Warburg effect is the observation made by Otto Warburg back in the 1920s of an increased glucose uptake and conversion to lactate in cancer cells [5]. He hypothesized that an irreversible alteration of mitochondrial respiration is at the origin of the development of cancer [6]. This hypothesis was ignored for more than fifty years by a large part of the scientific community, so much so that metabolism was not part of the “Hallmarks of cancer” listed in the famous review by Hanahan and Weinberg published in 2000 [74]. It was not until these were updated in 2011 that abnormal metabolism made its appearance with the mention of “deregulating cellular energetics” [71].

In honor of its discoverer, the activation of glycolysis under aerobic conditions has been dubbed the “Warburg effect”. At the same time that this process was admitted, it became a paradox to be resolved: why malignant cells, *a priori* presumed to have an increased need for ATP to proliferate, not use the OXPHOS pathway, despite its significantly higher energy yield?

Today, with the progress of knowledge on the origin and mechanisms of malignant transformation, Warburg’s hypothesis on the origin of uncontrolled cell proliferation is no longer accepted for two reasons. The first is that aerobic glycolysis is not directly related to malignancy. Many mammalian cell lines (lymphocytes and fibroblasts) develop aerobic glycolysis under proliferating conditions [75]. In cultured lymphocytes, after stimulation of proliferation, it has been shown that these cells convert a significant amount of consumed glucose into lactate [76,77,78]. The second reason is that the development of aerobic glycolysis may be independent of whether the mitochondria are functional or not. Indeed, several works have proven that mitochondria function normally in cancer cells and that blocking oxidative phosphorylation constitutes an adaptive event [79,80,81].

In addition, a study has investigated the impact of environmental acidity on tumor mitochondrial function [82]. The authors of this review reported that cancer cells reverted to OXPHOS when they were exposed to lactic acidosis, a common factor in tumor microenvironment. To draw this conclusion, they quantitatively determined the percentage of ATP production from glycolysis and OXPHOS with and without lactic acidosis. They found that in the absence of lactic acidosis, cancer cells exhibited excessive glycolysis and produced a large amount of lactate; while, in the presence of lactic acidosis, cancer cells showed a low level of glycolysis and produced a negligible amount of lactate. These quantitative data supported therefore their notion that cancer cells reverted from the Warburg effect to the OXPHOS phenotype under acidic conditions.

A reduced model of the cell energy metabolism was recently proposed to test the role of pyruvate in the distribution of fluxes toward respiration or fermentation, to specify the importance of lactate and to include the glycolytic inhibition by acidity [83]. The model is based on experimental data highlighting the role of acidity in the regulation of metabolism [84]. The results obtained from this model showed, with regards to the production of ATP, that keeping a constant level of pyruvate was necessary to direct the glycolytic fluxes toward fermentation when this exceeded the maximum activity of the pyruvate dehydrogenase (PDH) complex. Conversely, when the glycolytic flow is reduced by acidity, it is necessary to consume lactate. Thus, the oxidative activity is maintained as long as the oxygen level allows it and gradually passes to fermentation as long as the acidity is not too high, a transition dictated by the maintenance of the pyruvate concentration. This model tends to show that the Warburg effect is not necessarily an inherent characteristic of the tumor cell but a spontaneous and transient adaptation mechanism to a disturbed environment [85].

Beyond its influence on the Warburg effect, acidity in the tumor microenvironment has many effects on the evolution of cancer and its progression to malignancy. It has also been proposed that exposure to chronic acidosis induces genome instability through chromosome breakages and translocations driving somatic evolution [86,87]. On the other hand, acidity can be cytotoxic, inhibit cancer cell proliferation, and promote stress response and cell apoptosis [88,89,90,91]. Collectively, acidosis has been described as a defining hallmark of the tumor microenvironment. However, the mechanisms by which cancer cells, immune cells, and blood vessels sense acidosis and respond to it are yet to be completely established but may be advantageous for a more comprehensive understanding of tumor biology. This depends on the ability to make measurements, which we will now discuss.

## 4. Evolution of the Methods for Intracellular pH Measurements

In this section, we present some methods that have been developed to measure the intracellular pH of cells for a range of biological applications. The focus is here to illustrate the techniques and the contexts in which they have been developed. We note that most of the methods have been developed in contexts unrelated to cancer. However, all of the techniques are transposable to measure the intracellular pH in some other cell types, including cancer cells.

Measurement of the intracellular pH is indeed very challenging. Especially if we intend to make measurements at the single cell level in a non-disruptive and less intrusive way so as to be able to monitor acute pH changes and pH evolutions on a longer time scale (several hours to days). Although changes in the pH_*i*_ had been observed much earlier, the oldest method of what might loosely be described as pH_*i*_ measurement was employed in 1912 by Michaelis and Davidoff with red blood cells (for a review, see [92]). Using platinum/hydrogen electrodes, they noted that red cell lysis caused a change in bulk pH. Although cell lysis was the most appropriate method for estimating pH_*i*_, the potentially serious problems with this technique were recognized early on. For example, the metabolism continues after cell lysis and leads to a drop in pH due to the production of lactic acid and carbon dioxide. Therefore, various studies in the field of intracellular pH have been characterized by better developments in measurement techniques, by increasing knowledge of the mechanism of control of pH_*i*_, and by progress in establishing relationships between pH_*i*_ and cellular metabolism.

### 4.1. pH Microelectrodes

One of the methods developed to measure pH_*i*_ is the pH-sensitive microelectrodes method [93]. Although this method is technically very exacting, it has provided very important information. It is particularly suitable for providing direct, continuous monitoring of local pH and has been used extensively to measure cytosolic pH.

Due to the existence of electrical potential differences across the cell membrane, two electrodes must enter the cell to measure the pH_*i*_: a reference microelectrode filled with potassium chloride (KCl) to measure the membrane potential and a pH-sensitive microelectrode to measure a combination of pH and membrane potential. By subtracting the potential recorded by the two electrodes, one obtains an electric potential proportional to the local pH.

Using this method, intracellular pH measurement was performed in papillary muscle cells of streptozotocin-induced diabetic rats [94]. The aim of this study was to investigate the in vivo regulation of pH_*i*_ following an intracellular acid or alkaline loading in diabetic rats. Accordingly, diabetes was induced in male Wistar rats by the injection of streptozotocin into the femoral vein. The results show differences between diabetic and normal muscles in the regulation of pH_*i*_ in response to induced acidity. This suggests that diabetes is associated with a change in the activity of the Na^+^/H^+^ exchange [95], which is a major pathway for the export of H^+^ ions from cells during acidity [25].

Further, pH-sensitive microelectrodes have been used to measure the pH_*i*_ of mammalian cardiac cells. The first measurement in vivo was performed by Ellis and Thomas, (1976) [96] who studied the effect of CO_2_ on the pH_*i*_ of Purkinje fiber cells cut from sheep ventricles. The results showed that the pH_*i*_ was strongly influenced by CO_2_ in the equilibration gas. The mean pH_*i*_ in the presence of CO_2_ is close to 7.02 and is significantly lower than when CO_2_ is absent.

In fact, when the dimensions of the cells are too small, such as epithelial cells, it is difficult and often impossible to accommodate two separate electrodes in the same cell. In this case, two different cells can be penetrated and a distinct membrane electrical potential can be noted by the reference microelectrode and the pH-sensitive microelectrode. The measurement of pH_*i*_ is therefore carried out by calculating the average values of the electrical potentials of several penetrations [97]. In order to circumvent most of these problems, a double-barreled pH microelectrode was introduced [98]. It allows the penetration of both electrodes into the same cell, thus measuring the same membrane potential.

A repetitive pH_*i*_ measurement of Purkinje fiber cells was obtained using the double-barreled pH microelectrode [98]. The results show that the pH_*i*_ varies between 7.1 and 7.2, which is, however, significantly greater than that reported in [96]. It has been reported that this difference is probably due to the different ${\mathrm{HCO}}_{3}^{-}$ concentration in the control solution, compared to that used in [96].

Hagberg et al. (1983) [99] measured in vivo the pH_*i*_ of rabbit skeletal muscle fibers using the double-barreled pH microelectrode. The pH_*i*_ of 7.00 ± 0.09 found in this study was lower than that found in mouse soleus fibers in vitro using pH-sensitive microelectrodes (7.07) [100] or in rat soleus using double-barreled electrodes (7.1–7.2) [98]. They assumed that the higher bath pH and the lower level of CO_2_ in those studies could explain this small discrepancy. In addition, the pH_*i*_ could be different in the soleus (composed mainly of red fibers) than in the predominantly white gastrocnemius muscle.

Despite the widespread use of pH-sensitive microelectrodes for pH_*i*_ measurements, their use has been rather limited because they can cause cell membrane damage and cytosolic leakage that alters pH_*i*_ measurements. In addition, the use of pH-sensitive microelectrodes is impossible on mobile cells such as amoebae [101].

### 4.2. *^31^*P NMR Spectroscopy

Following the pioneering work of Moon and Richards [102], who, in 1973, used the principle of the variation in the chemical shift of mainly cytosolic inorganic phosphate (P_*i*_) with pH, phosphorus nuclear magnetic resonance (^31^P NMR) spectroscopy was used as a non-invasive pH_*i*_ measurement technique. In this case, the cells are incubated in the presence of probes, such as phosphate or methylphosphonate comprising ^31^P, which penetrate into the cells.

Several cardiac pH measurements were performed using ^31^P NMR spectroscopy [103,104,105,106]. Since cardiac cellular acidosis has been introduced as one of the central mechanisms to explain irreversible cell damage during ischemia [107], cardiac cellular pH is therefore an important parameter to measure in various types of ischemic states. For example, a ^31^P NMR measurement of cardiac pH was performed in an isolated rat heart during ischemia and reperfusion [105]. The results show that in the pre-ischemic rat heart, the pH_*i*_ was approximately 7.15. After 4 min of ischemia, the pH_*i*_ decreased to approximately 6.16. On reperfusion, the pH_*i*_ returned to pre-ischemic levels within 2.5 min. In addition, ^31^P NMR spectroscopy has also been used to monitor renal intracellular pH [108,109,110]. It has been suggested that intracellular acidification could be a primary signal that stimulates renal ammoniogenesis [111]. Therefore, knowing renal intracellular pH may be a key issue in understanding the regulation of ammoniogenesis. In this context, the intrarenal pH was measured in vivo in perfused rat kidneys where an acidification, from pH 7.4 to pH 6.9, was induced by the addition of hydrochloric acid (HCl) to the perfusion medium [109]. The results showed that the kidney pH fell from 7.20 to 6.75 and reached a plateau of pH 6.85, 30 min after addition of HCl. In parallel, it was found that the rate of ammonia (NH_3_) production at 7.4 was doubled when the perfusate was acidified to pH 6.9. This is therefore consistent with the hypothesis that intracellular acidification may participate in the stimulation of ammoniogenesis.

In addition to measuring intracellular pH, ^31^P NMR spectroscopy has also been used to visualize and quantify the phosphorylated metabolites, which are directly involved in energy metabolism. We will cite, as examples, ATP which constitutes the main source of energy of the organism; phosphocreatine, which is an energy buffer (It makes it possible to maintain the concentration of available ATP constant.); and the inorganic phosphate P_*i*_, which is the substrate for the ATP synthesis reaction. In addition to quantification, ^31^P spectra provide unique access to dynamic parameters, such as the speed of ATP synthesis [112,113]. Although this method can provide very important information, it remains little used. This is due to relatively high equipment costs and the need to work with suspensions of high cell concentrations. In addition, it has the disadvantage of giving relatively long analysis times (spectrum acquisition), which do not make it possible to see rapid variations in the pH_*i*_ [114,115,116].

### 4.3. Weak Acids and Bases

The weak acids and bases distribution method has also been introduced to measure intracellular pH [117]. The basis of this method is the use of weak acids HA when pH_*i*_ > pH_*e*_ or of weak bases B when pH_*i*_ < pH_*e*_. The neutral species of these probes diffuses freely through the membrane while the ion is impermeable. The weak protonated acid, HA, transports its proton across the membrane and can then dissociate inside the cell, lowering the intracellular pH and increasing the extracellular pH (pH_*e*_). The weak deprotonated base B will penetrate the cell, leaving a proton behind, and will tend to pick up a proton inside the cell, raising the pH_*i*_ and lowering the pH_*e*_ (Figure 4).

In the case of a weak acid, [HA]_*i*_ = [HA]_*e*_ at equilibrium. Moreover, since the acid dissociates on both sides, it is therefore assumed that the dissociation constant is the same in the extracellular and intracellular media. From this equilibrium, the following equation results:(1)$$K_{a}=\frac{{\left[H^{+}\right]}_{i}{\left[A^{-}\right]}_{i}}{{\left[HA\right]}_{i}}=\frac{{\left[H^{+}\right]}_{e}{\left[A^{-}\right]}_{e}}{{\left[HA\right]}_{e}}$$

Under this condition, and if pH_*e*_ > pK_*a*_ + 1, most of the acid is ionized on both sides of the membrane, and the measurement of the acid distribution ratio allows calculation of the intracellular pH at equilibrium. Since the extracellular pH can be easily measured, the calculation of the pH_*i*_ then becomes possible. The quantities of ionized acid are assayed by titrimetry or radiometry in the case of labeled probes, such as acetate and methylamine [118].

Waddell and Butler (1959) [119] measured the pH_*i*_ of the skeletal muscle of the dog using the weak acid DMO (5,5-Dimethyl-2,4-oxazolidinedione). The effects of various treatments altering the pH or the pCO_2_ (plasma CO_2_) of plasma were studied. In 11 dogs, the calculated pH_*i*_ values of normal, resting muscle were all in the range from 6.96 to 7.10 with an average of 7.04. The greatest changes in muscle pH_*i*_ are those produced by increasing blood carbon dioxide tension. The pH_*i*_ of muscle is lowered by a high tension of carbon dioxide whether or not the blood pH is lowered. Lowering blood pH by intravenous infusion of hydrochloric acid (HCl) has less effect on lowering the muscle pH_*i*_ than the high tension of carbon dioxide. While raising blood pH, sodium bicarbonate (NaHCO_3_) injection raises the muscle pH_*i*_ only to a small extent.

Although this technique is easy, accurate and applicable to small cells, its use is limited by the fact that it requires the destruction of the tissue and therefore cannot easily be used to measure pH_*i*_ changes over time. In addition, the presence of acidic or basic molecules can disrupt the establishment of the electrochemical gradient [116].

### 4.4. Fluorescence Microscopy

Another method developed over the past thirty years to deal with these drawbacks is fluorescence microscopy. By the protonation of a chemical function, it is possible to modify the spectroscopic properties of the molecule and to obtain a probe sensitive to the variation in the proton concentration.

Currently, pH-sensitive probes are often in the esterified form of the probe (AM). The presence of these ester groups facilitates the diffusion of fluorophores through the cytoplasmic membrane. Inside the cells and under the action of intracellular esterases, the ester groups are cleaved and the more or less negatively charged probes are retained intracellularly and are then potentially fluorescent.

Gores et al. (1989) [120] used the BCECF-AM probe to measure the pH_*i*_ in cultured rat hepatocytes after ATP depletion by metabolic inhibition with KCN (potassium cyanide) and iodoacetate acid (IAA)(chemical hypoxia). KCN and IAA block cellular ATP formation through inhibition of oxidative phosphorylation and glycolysis, respectively. During chemical hypoxia at a pH_*e*_ of 7.4, the pH_*i*_ dropped from 7.36 to 6.33 within 10 min and remained at 6.1–6.5 for 30–40 min (plateau phase). Subsequently, the pH_*i*_ began to increase, and cells ensued within minutes, as evidenced by nuclear staining with propidium iodide. Exposure of hepatocytes to an acidic pH_*e*_ simultaneously with KCN and IAA produced a slightly greater drop in the pH_*i*_, prolonged the plateau phase of intracellular acidosis, and delayed the onset of cell death. In addition, inhibition of the Na^+^/H^+^ exchange with amiloride also produced an increase in cell survival and a prolongation of the plateau phase of intracellular acidosis. The results suggest that intracellular acidosis after ATP depletion protects against hepatocellular death from ATP depletion, a phenomenon that may represent a protective adaptation against hypoxia-induced stress [120].

It has been suggested that such an alteration in the pH_*i*_ of brain cells could contribute to neurological dysfunctions in patients with AIDS. Therefore, Makutonina et al. (1996) [121] used the BCECF-AM probe to quantitate the pH_*i*_ in HIV-1-infected T cells (human immunodeficiency virus type 1). The results obtained show that an infection of RH9 cells, an established line of ${\mathrm{CD}}_{4}^{+}$ T-lymphoblastoid cells, with HIV-1, resulted in a significant decrease in the pH_*i*_ from 7.2 in mock-infected cells to approximately 6.6 by day 4 after the infection when cells were undergoing acute cytopathic effects. Fluorescence concentration analysis using BCECF indicated that the pH_*i*_ in persistently infected cells that survived the acute cytopathic effects of HIV-1 was from approximately 6.8 to 7.0, values which are modestly lower than those in mock-infected cells. The mechanism by which HIV reduces the pH_*i*_ in ${\mathrm{CD}}_{4}^{+}$ T-lymphoblastoid cells is not established in this study. However, prior studies indicated that HIV infection induces intracellular acidification in lymphocytes due to dysfunction of the plasma membrane ion transport system Na^+^/H^+^ [122,123]. Exposure to gp120, the HIV-1 surface glycoprotein (SU), activates the Na^+^/H^+^ exchange system, resulting in an increase in the active transport of protons out of the cell and intracellular alkalinization. The results of these studies suggest that alterations in the pH_*i*_ may mediate certain cytopathic effects of HIV-1, thereby contributing to depletion of the ${\mathrm{CD}}_{4}^{+}$ T lymphocytes in patients with AIDS.

Salvi et al. (2002) [124] evaluated the performance of three pH-sensitive probes, SNARF-1-AM, BCECF-AM, and Cell-Tracker Green CMFDA, in Chinese hamster ovary (CHO), human embryonic kidney (HEK293), and two human colon carcinoma (Caco-2) cells to measure pH_*i*_ changes in a multi-well plate format. The stability of the pH_*i*_ measurements was strongly affected by the rate of the probe efflux, which was found to depend on the pH-sensitive probe, temperature, pH_*e*_, cell type, and the use of transport inhibitors to prevent probe export. The results show that the performance of the Cell-Tracker Green CMFDA was significantly better with respect to the cellular probe efflux. Therefore, this probe was selected for use in further studies with CHO, HEK293, and Caco-2 cells. To investigate whether the effect of hypertonic stress on the pH_*i*_ can be detected, the medium was made hypertonic by the addition of sucrose. The exposure of HEK293 and CHO cells to the hypertonic medium induced pH_*i*_ increases of 0.08 pH unit and 0.16 pH unit, respectively, whereas no significant pH_*i*_ rise was observed in the Caco-2 cells. To examine whether the activation of G-protein coupled receptors was connected to a pH_*i*_ change, acetylcholine was injected into HEK293-M1 and CHO-M1 cells. In both cell lines expressing a high level of muscarinic receptors, activation by acetylcholine did not cause any significant pH_*i*_ response compared with wild-type cells. Similarly, stimulation of endogenously expressed *β*-adrenergic receptors by isoproterenol had no effect. The effects of human dipeptide transport activity on the pH_*i*_ were also investigated in this study. A pH_*i*_ decrease of 0.15 pH unit was observed in CHO cells expressing the human H^+^/peptide transporter PEPT1 upon addition of dipeptide substrates. The authors of this study succeeded in developing a rapid fluorimetric assay using the multi-well plate format to measure pH_*i*_ changes in living cells and examine pH_*i*_ changes induced by various effectors.

A large number of fluorescent probes, or fluorophores, have been developed for pH_*i*_ sensing; these probes have the advantage of low toxicity to cells and are not metabolized [116]. In addition, the specificity of localization can allow the exclusive study of certain cellular compartments, such as lysosomes in macrophages [125]. Since its introduction by Tsien in the early 1980s, 2′,7′ bis(carboxyethyl)-5(and 6)-carboxyfluorescein (BCECF) has been by far the most widely used pH-sensitive fluorescent probe [126].

In a recent study, we developed a methodology that allowed us to improve the precision of pH measurements made with the BCECF probe and to extend the pH range that can be measured with this probe. We validated this methodology by measuring the pH drift observed in the cell culture medium [127]. We then applied it to monitor simultaneously the intra and extracellular pH in two glioma cell lines (murine F98 and human U87 gliomas) that were submitted to a range of acute pH changes from 5.0 to 8.4 [19]. We showed that the two cell lines reacted differently to acidity. The F98 cells exhibited a weak pH_*i*_ regulation capability, whereas the U87 cells were more efficient at raising their pH_*i*_ under acidic environmental conditions. On the other hand, the F98 cells exhibited a higher survival rate compared to that of the U87 cells under those conditions. We hypothesized that the F98 cells were activating the V-ATPase pump to encapsulate the protons in the lysosomes to neutralize the excess acidity and keep functioning. Although this mechanism is well known, it remains to be verified in this specific case.

## 5. Acidity and Therapy

Until very recently, the emphasis in anti-cancer drug research and development solely targeted tumor cells. Conventional chemotherapy primarily aimed at hindering cancer cell proliferation, while more recent targeted agents focused on specific tumor proteins or neovascularization. However, with minimal prognosis improvement and a growing understanding of acidity’s role in tumor progression and treatment response, significant attention has shifted toward strategies addressing tumor acidity. These strategies, which directly target acidity or leverage its characteristics, are now under development and show promising initial results. For example, some new strategies focus on exploiting acidity to improve drug uptake by the tumor cells, to activate the drugs, or to enhance their toxic effects. Another approach involves inhibition of the proteins responsible for maintaining an intracellular pH, vital for tumor survival in acidic conditions. This is achieved through the use of small molecule inhibitors or antibodies targeting these proteins.

### 5.1. Extracellular Acidity and Chemotherapy

Acidosis of the tumor microenvironment leads to a selection of cancer cells with a stem cell phenotype, with greater invasive and metastatic potentials, and failure with treatment strategies involving chemotherapy [67,128,129,130]. The association between extracellular acidity and increased chemotherapeutic resistance has been reported in previous in vitro and in vivo studies on human melanoma [131], osteosarcoma [132], prostate carcinoma [133], colon [134], breast [135], and human oral squamous cell carcinoma [136]. Therefore, the following paragraph is devoted to a literature review in order to elucidate the biological mechanisms underlying the effects of acidosis on chemotherapies (Table 1).

The passage of chemotherapeutic agents through the cell membrane is often compromised by the nature of the molecule. Uncharged and small molecules can easily diffuse whereas the passage of charged or bigger molecules can be much more difficult. ATP-Binding Cassette (ABC) transporters, also known as primary active transporters, are a broad family of proteins involved in the transport of hydrophobic compounds against their concentration gradients, from a wide variety of substrates, including peptides, lipids, and chemotherapeutic agents [137,138]. ABC transporters use the energy provided by the hydrolysis of the ATP at the level of NBDs (nucleotide-binding domains) to function. Genes encoding ABC transporters are grouped into subfamilies according to their domain organization and amino acid homology. In humans, seven subfamilies have been identified (from ABCA to ABCG).

Cheng et al. (2012) [134] assessed in vitro the regulation of the ABCG2 transporter in human colon carcinoma, HCT-116, and S1 and its resistant S1M1-80 cell lines overexpressing ABCG2 in response to an acidic pH. The data revealed increased regulation of ABCG2 gene transcription, with the exception of the S1 cell line, under extracellular acidity, and increased resistance to the drugs Cisplatin (CIS) and mitoxantrone [134].

Federici et al. (2014) [139] have investigated the role of both extracellular acidosis and exosome release in resistance of different malignant cell lines (human breast cancer, human metastatic melanoma, and human colon carcinoma) to Cisplatin. They demonstrated that tumor cell lines exhibited different sensitivity to the Cisplatin and that the acid culture condition reduced sensitivity to Cisplatin in all tumor cell lines tested compared to the same cells cultured at a neutral pH. In addition, the results showed that an acidic pH increased exosome release by tumor cells with a higher accumulation of Cisplatin, which may be one of the mechanisms by which cells protect themselves by expelling the drug. Melanoma cells’ pre-treatment with lansoprazole, a proton pump inhibitor (PPI), induced a 50% reduction in the amount of Cisplatin present in the exosomes purified from the cell culture supernatant, as compared to the exosomes purified from the supernatant of the cell cultures that were not treated with lansoprazole. This supported the hypothesis that pretreatment with a proton pump inhibitor may lead both to exosome release inhibition and an increased drug retention by tumor cells [139].

The most important human ABC transporter involved in drug disposition is ABCB1, also known as multidrug resistance protein 1 (MDR1) or P-glycoprotein (Pgp) [140]. P-glycoprotein prevents cellular uptake of a large number of structurally and functionally diverse compounds by acting as an efflux pump and transporting the chemotherapeutics out of the cell and, in this way, causes multidrug resistance (MDR) [141,142].

A group of researchers have been working on the impact of extracellular acidity on the activity of Pgp and the cytotoxicity of chemotherapeutic drugs [133]. They found that exposing prostate carcinoma cells (AT1) to an acidic extracellular environment (pH 6.6) for a period from 3 to 6 h doubled the activity of Pgp and reduced the cytotoxic efficacy of two chemotherapeutic drugs, Daunorubicin (DNR) and Cisplatin. In order to assess the cytotoxic activity of the chemotherapeutic drugs, the relative caspase 3 (a proapoptotic protein) activity normalized with respect to untreated control cells was measured. With Cisplatin, caspase 3 activity was significantly reduced by the acidic environment compared to the normal pH. However, even though Cisplatin-induced caspase 3 activation was markedly reduced in the acidic microenvironment, overall cell death was not significantly different in cells incubated with Cisplatin at different pH levels. These results indicate that the cytotoxicity of Cisplatin was only marginally affected by the acidic microenvironment. In contrast, DNR-induced cell death was markedly reduced by the acidic microenvironment. In order to investigate the underlying mechanisms by which the extracellular pH may influence pGP activity, changes in the intracellular levels of calcium ions (${\mathrm{Ca}}^{2+}$) and protein kinase C (PKC) were measured. It was found that the increased Pgp activity under acidic conditions was related to the decreased intracellular calcium levels and inhibition of PKC. Based on the results of these experiments, the authors concluded that the chemoresistance to chemotherapeutic agents generated by extracellular acidosis may be the result of an increase in Pgp activity, which, in turn, is potentially caused by a decreased intracellular calcium concentration and PKC activity (The decreased intracellular calcium concentration can also be induced by reducing the extracellular one which leads to the same effect as acidosis) [133].

Thews et al. (2014) [143] studied the cytotoxicity of three chemotherapeutic drugs, DNR, CIS, and docetaxel (DOC), and, their dependence on Pgp activity during acidosis was analyzed. In vitro and in vivo experiments were performed using the same treatments and the same cell lines as in their previous study. Under acidic (pH 6.6) conditions, they observed a decrease in the cytotoxicity of DNR or DOC, whereas Cisplatin-induced cell death was almost pH-independent. The Pgp inhibitor (verapamil) reversed the acidosis-induced chemoresistance against DNR and DOC. The cytotoxicity in vivo was assessed by forcing glycolytic metabolism “acidosis treatment”) in the tumor tissue. In vivo results confirmed that induction of acidity in animals reduced the cytotoxicity of DNR and DOC, whereas the cytotoxicity of CIS remained almost independent from the tumor pH. Thus, it was suggested that when there are clinical indications for DAU and DOC treatment, the Pgp-mediated chemoresistance can be counteracted by inhibition of the drug transporter [143].

**Table 1 biology-13-00225-t001:** Literature review on the effects of pH on chemotherapies.

Reference	Cancer Type	Cell Model	Tested Drug	pH Assessed	Mechanisms Evaluated	Main Results
Mellor and Callaghan (2011) [144]	Colorectal adenocarcinoma (DLD-1), colon adenocarcinoma (HT29), and ovarian adenocarcinoma (NCI^*ADR*^)	in vitro	Doxorubicin (DOX)	6.9–7.0 and 7.2–7.3	Growth of tumor spheroids in normoxia and hypoxia, intracellular accumulation of DOX, and inhibition of Pgp by Tariquidar (XR9576)	The distribution and accumulation of DOX were heterogeneous in all cell lines evaluated. The acidity generated by hypoxia decreased the accumulation of DOX in tumor spheroid. The inibition of Pgp by Tariquidar (XR9576) increased the accumulation of DOX in tumor spheroids.
Avnet et al. (2016) [132]	Osteosarcoma	in vivo	Doxorubicin (DOX)	6.5 and 7.4	Combined treatment of DOX and omeprazole (a proton pump inhibitor targeting lysosomal acidity)	The combined treatment of DOX with omeprazole showed a higher necrotic areas, smaller tumor volumes, and less body weight loss.
Fan et al. (2012) [135]	Breast cancer (MCF-7/ADR)	in vitro and in vivo	Doxorubicin (DOX), Cisplatin (CIS), and 5-fluorouracil (5-FU)	Acidic pH (unknown value)	Induction of LASS2 expression	The overexpression of LASS2 in MCF-7/ADR breast cancer cells increased the effect of several chemotherapeutic agents. LASS2 inhibited the function of V-ATPase. More DOX entered the cells and stayed in the nuclei of cells, inducing increased rates of apoptosis.
Visioli et al. 2014 [136]	Endothelial cells from human oral squamous cell carcinomas (OSCC)	in vitro	Sunitinib	6.0–6.4	Activation of UPR: quantification of Grp78 in endothelial cells and cytotoxicity using SRB	Extracellular acidity increased expression of the UPR marker (Grp78) and its inhibition reversed the drug sensitivity to Sunitinib.
Cheng and To (2012) [134]	Human colon carcinoma HCT-116 and S1 and its ABCG2-overexpressing resistant S1M1-80 cell lines	in vitro	Cisplatin (CIS) and mitoxantrone	5.00	Regulation of ABCG2 under adverse conditions within the tumor microenvironment (hypoxia, glucose deprivation, and acidosis)	Glucose depletion, decreased extracellular pH, and hypoxia can all upregulate ABCG2 transcript level leading to multidrug resistance. Acidic pH did not significantly alter the level of ABCG2 in S1 cells.
Federici et al. 2014 [139]	MCF7 (human breast cancer), Me30966 and Me501 (human metastatic melanoma), and SW480 (human colon carcinoma) cell lines	both in vitro and in vivo experiments	Cisplatin (CIS)	acidic (5.0–6.0), buffered (7.4), and unbuffered media	Cisplatin cellular resistance and effects of proton pump inhibitor (lansoprazole) on Cisplatin tumor uptake	Treatment with lansoprazole increased the intracellular absorption of Cisplatin and reduced the amount of Cisplatin present in the exosomes.
Thews et al. (2006) [133]	Rat prostate cancer (AT1)	in vitro	Daunorubicin (DNR) and Cisplatin (CIS)	6.6 and 7.4	Caspase 3 activity and cell toxicity assay	Exposure to acidic extracellular environment doubled Pgp activity and reduced cytotoxic efficacy of CIS and DNR.
Thews et al. (2014) [143]	Rat prostate cancer (AT1)	in vitro and in vivo	Daunorubicin (DNR), Cisplatin (CIS), and docetaxel (DOC)	6.6 and 7.4	Measurement of apoptosis induction (caspase 3) and cell survival	Acidity reduced the cytotoxicity of DAU, CIS, and DOC. The Pgp inhibitor (verapamil) reversed the acidosis-induced chemoresistance against DNR and DOC.

In an acidic microenvironment, combined chemotherapy and proton pump inhibitors have been shown to increase the sensitivity of cancer cells to treatment. Avnet et al. (2016) [132] injected osteosarcoma cells into immunodeficient mice (NOD/SCID animals) and treated them with the chemotherapeutic drug DOX and omeprazole, a proton pump inhibitor targeting lysosomal acidity. Exposure to an acidic pH_*e*_ significantly increased the number and acidity of lysosomes, and the combination of DOX with omeprazole significantly enhanced DOX cytotoxicity. In addition, the results of this study demonstrated that mice treated with the combination of DOX and omeprazole showed a higher necrotic index, smaller tumor volumes, and less body weight loss compared to the groups treated with DOX alone. Therefore, it can be inferred that proton pump inhibitor drugs may offer novel solutions to overcoming drug resistance.

The role of V-ATPase in the efficacy of chemotherapy treatment has been evaluated in breast cancer. Fan et al. (2012) [135] observed that LASS2 (Homo sapiens longevity assurance homolog 2 of yeast LAG1) is able to enhance breast cancer chemosensitivity by counteracting the acidic tumor microenvironment by inhibiting proton pump V-ATPase activity. Low expression of LASS2 was associated with poor prognosis in breast tumors. The results showed that the overexpression of LASS2 in MCF-7/ADR breast cancer cells increased the effect of several chemotherapeutic agents (DOX, CIS, and 5-fluorouracil (5-FU)). In these cells, they detected that LASS2 inhibited the function of V-ATPase by binding to its C-subunit. This effect was mediated by a significant increase in pH_*e*_ and lysosomal pH, and DOX that was previously sequestered in the cytosol was released in the nuclei of cells, inducing increased rates of apoptosis. These results suggest that LASS2 is involved in chemotherapeutic outcomes and that a low expression of LASS2 may predict drug resistance.

Visioli et al. (2014) [136] investigated the effect of acidic pH stress on tumor-associated endothelial cells (TECs) and found that the mechanisms underlying cell adaptation and resistance to antiangiogenic chemotherapeutic agents may depend on the activation of the unfolded protein response (UPR). Primary human dermal microvascular endothelial cells (HDMECs) cultured in an acidic-conditioned medium showed increased expression of the UPR marker (Grp78, glucose-regulated protein 78) and concomitantly greater resistance to cytotoxicity induced by the drug Sunitinib. Inhibition of GRP78 resensitizes HDMECs to drug treatment. Therefore, this work suggested that UPR induction in endothelial cells under acidic pH conditions may results in therapeutic failures in response to chemotherapy. Targeting Grp78, the key component of the UPR pathway, may provide a promising alternative for cancer treatment.

Tumor spheroids have also found application in assessing tumor resistance since they reproduce the tumor microenvironment conditions in a more accurate way and provide more physiologically relevant information and more predictive data for in vivo tests. Mellor and Callaghan, (2011) [144] used such a model to evaluate chemoresistance in colorectal adenocarcinoma cells (DLD-1), colon adenocarcinoma cells (HT29), and ovarian adenocarcinoma cells (NCI^*ADR*^). The accumulation and distribution of doxorubicin (DOX), a chemotherapeutic drug, were measured in tumor spheroids. DOX is fluorescent and thus allows detection by confocal microscopy. Tumor spheroids demonstrated a high level of DOX in cells on the tissue surface. An inner ring, only in the DLD-1 and HT29 cell lines, consistent with the hypoxic zone, was almost completely devoid of DOX staining, reflecting poor intracellular accumulation in this zone. In addition, the authors of this study observed that the pH was acidic (6.9–7.0) in all the cell lines exposed to hypoxia. The influence of Pgp on doxorubicin accumulation was examined under hypoxic conditions in TS comprising all three cell types. The inhibition of Pgp by the modulator Tariquidar (XR9576) increased the accumulation of DOX in all three cell lines evaluated. The study concluded that exposure of 3D tumor spheroids to hypoxia generates an acidic microenvironment; this leads to increased expression of drug transporters and, therefore, a decrease in DOX-induced cytotoxicity [144].

### 5.2. The Case of Temozolomide

Temozolomide (TMZ) is a DNA alkylating agent discovered in the early 1980s that is now a reference in the treatment of glioblastomas. Temozolomide has the ability to cross the blood–brain barrier and has excellent biodistribution in the central nervous system as well as high stability at acidic pH, allowing its oral administration. The activation of this molecule takes place via the spontaneous conversion of temozolomide into 3-methyl-(triazen-1-yl) imidazole-4-carboxamide (MTIC), which is subsequently degraded into a methyldiazonium cation, the DNA-methylating species. The methyldiazonium cation activates the DNA repair system MMR (mismatch repair), which will lead to DNA breaks and trigger cell death via the p53 protein [145,146,147].

The degradation rates of both TMZ and MTIC exhibit strong opposite pH dependencies, with an exponential increase and decrease observed as pH values vary, respectively. Consequently, only a narrow pH range facilitates the hydrolysis of TMZ into MTIC and, subsequently, into the DNA alkylating group [145]. Stéphanou and Ballesta (2019) [148] have designed a theoretical framework to study the antitumor efficacy of TMZ, integrating the effect of the pH_*e*_, the spatial configuration of the tumor, and the microenvironment in order to identify this range. Based on the relationship between the pH_*e*_ and pH_*i*_ for a given cell type, it provides quantitative predictions regarding the differential impact of TMZ on cancer cells and optimal pH values leading to a greater amount of DNA damage. Therefore, by experimentally characterizing the cell’s ability to regulate its pH_*i*_ as a function of the pH_*e*_, the model highlights potential ways to improve the efficacy of TMZ on tumor cells. We subsequently characterized the pH regulation ability of two glioma cell lines and were indeed able to show that different optimal pHs for TMZ efficacy were identified [19].

Despite the treatment’s promise, the use of TMZ as an oral chemotherapy agent has certain limitations. Unfortunately, more than 50% of GBM patients treated with TMZ do not respond to therapy [149,150]. In addition, some patients subsequently suffer from recurrences or their tumor continues to grow. This represents treatment failure that is commonly associated with the phenomenon known as drug resistance, which is a major problem with TMZ treatment. This calls for urgent means to circumvent those limitations.

### 5.3. Targeting the Membrane Transporters

The activity of NHE-1 participates in the phenomena of resistance to anti-cancer drugs. Cells resistant to chemotherapy agents generally have a more alkaline cytoplasmic pH and more acidic intracellular vesicles than cells sensitive to these agents [151]. It has been shown that the expression and activity of NHE-1 were increased in cells resistant to doxorubicin and that inhibition of NHE-1 by a pharmacological inhibitor restores sensitivity to this anti-cancer agent [152]. Inhibition of NHE-1 is also involved in the action of certain anti-cancer agents that induce cancer cell apoptosis such as paclitaxel [11] and Cisplatin [153].

Targeting V-ATPase activity has been proposed as a new cancer treatment strategy [154]. A V-ATPase inhibitor molecule, esomeprazole used as an acidic pH-activated prodrug, decreases tumor growth and tumor pH gradient in vivo [155]. V-ATPase inhibitors could also be used in the context of resistance to chemotherapy. Indeed, the expression of several V-ATPase subunits is increased by a chemotherapy agent, Cisplatin [156,157]. This increase in expression is involved in chemoresistance phenomena; therefore, it has been proposed that inhibition of V-ATPase would increase the cytotoxicity of Cisplatin and could be a new strategy for avoiding resistance to this compound.

The use of an MCT1 inhibitor, *α*-cyano-4-hydroxycinnamate (CHC), has been proposed in order to inhibit the entry of lactate into oxidative cells, which will then have to use glucose as an energy substrate and which will not be, therefore, more available to the glycolytic cells in the center of the tumor [35]. The use of CHC shows a decrease in tumor growth and radiosensitization in mouse tumor models [35]. Similar effects are found when the expression of MCT1 is inhibited by an siRNA [158]. Other inhibitors have been identified [159,160]. No specific MCT4 molecule is available, and a targeting problem may arise since MCT4 is located in glycolytic cells far from blood vessels.

CAIX represents a particularly interesting therapeutic target and is the subject of intense research around the development of specific inhibitors [161].

## 6. Conclusions

Normal and tumor cells share a fundamental similarity in their regulation of intracellular pH, both striving to maintain it near the physiological levels necessary for cellular function. However, chronic adverse conditions within the tumor microenvironment favor the selection of resistant, invasive, and aggressive cell populations. These cells develop enhanced abilities to withstand, manage, or regulate acidity, in contrast to normal cells that may succumb to such conditions and perish.

While many experiments have investigated cells within their native environments, understanding their intrinsic functioning often requires exposure to controlled conditions outside the norm, for instance, exploring the dynamic capacity of cells to regulate the intracellular pH necessitates subjecting them to a broad range of pH levels, from highly acidic to basic, in a systematic manner. Our research with two glioma cell lines demonstrated the potential to optimize temozolomide efficacy through this approach [19], although it has faced criticism for exposing cells to extreme conditions not typically encountered in vivo.

A significant challenge in such experiments lies in simultaneously measuring the extracellular and intracellular pH levels in individual cells over extended periods, spanning from hours to days. Fluorescence microscopy employing pH-sensitive probes offers a viable alternative, though protocols must be carefully adjusted to prevent cell overexposure to light excitation or signal extinction within tumor masses.

This review, spanning the past century, highlights a persistent interest in acidity regulation and measurement. Today, there is a renewed interest in understanding the role of acidity in driving cellular progression toward malignancy and increased aggressiveness, as well as its potential implications for therapy.

The exploration of acidity regulation and measurement underscores the critical role of the acidic tumor microenvironment in driving cancer progression and therapy resistance. Understanding how cells respond to varying pH conditions provides valuable insights into the development of targeted cancer therapies. By leveraging this knowledge, innovative approaches can be designed to disrupt tumor survival pathways and enhance treatment efficacy. Strategies such as pH-modulating agents and proton pump inhibitors offer promising avenues for combating acidity-related challenges in cancer treatment. By targeting acidity as a therapeutic vulnerability, personalized medicine approaches can be tailored to individual tumor characteristics, maximizing treatment effectiveness. Thus, the renewed interest in acidity as a therapeutic target aligns with the ongoing efforts to advance cancer treatment strategies and improve patient outcomes.

## Figures and Tables

**Figure 2 biology-13-00225-f002:**
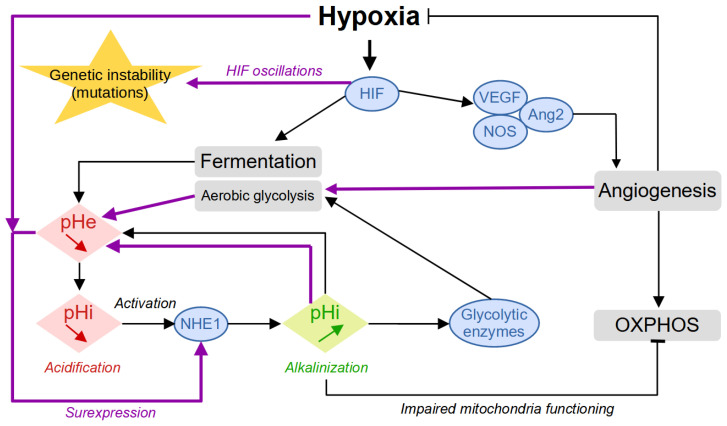
Building up acidity in the tumor microenvironment. The first reaction to hypoxia is represented with the black arrows. The secondary response, promoted by the worsening of the conditions with increased environmental acidity and intermittent oxygen levels induced by angiogenesis, is represented by the purple arrows.

**Figure 3 biology-13-00225-f003:**
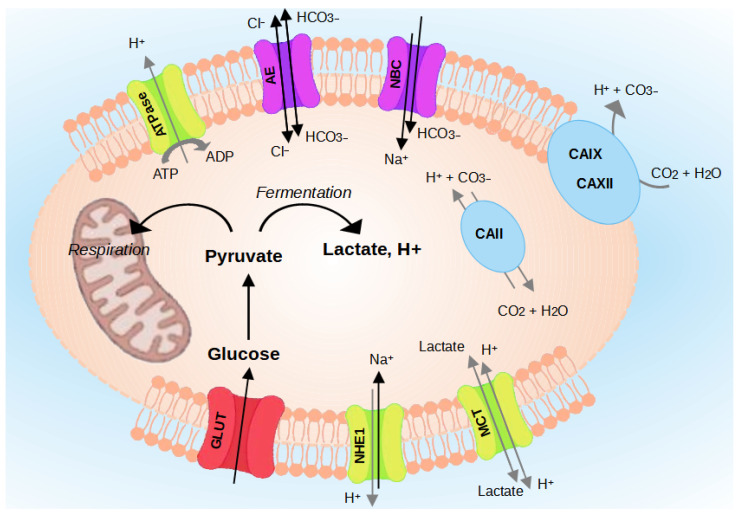
Regulation of intracellular pH (pH_*i*_) in cancer cells. Cancer cells survive in an acidic environment by coordinating the function of pH-regulating proteins, such as NHE1, CAs, MCTs, V-ATPase, and ${\mathrm{HCO}}_{3}^{-}$ transporters.

**Figure 4 biology-13-00225-f004:**
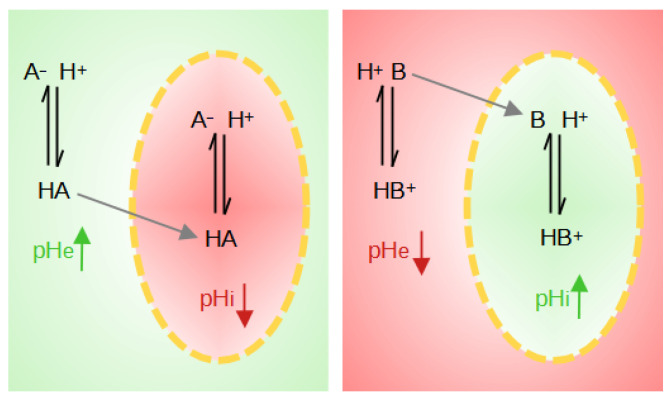
Illustration of the effects on local pH when weak acids (HA) or weak bases (B) are present. The yellow dotted line represents the cell membrane. The red zones correspond to a pH drop, whereas green zones correspond to a pH rise.

## Data Availability

Not applicable.

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
