# Peer review of "On the Importance of Acidity in Cancer Cells and Therapy"

_biology, 2024, doi:10.3390/biology13040225_

Round 1

Reviewer 1 Report

Comments and Suggestions for Authors

The work by A. Tafech and A. Stéphanou represents a comprehensive tutorial review of the current understanding of pH dependent biological processes.

One comment before the review can be recommended for publication: the title and the majority of the text address cancer biology whereas a sizeable part in the middle describes the methods and measurements in cancer-unrelated situations. Please either correct the title or edit the review to avoid distraction from the main focus.

Specific comments:

1.    The review addressed the problem of acidity and its regulation
mostly in tumor cells. This pivotal problem is not limited to cancer
or other eukaryote cells but is pertinent to biological regulation in
all species. The main questions are the role of pH in tumor biology
and the mechanisms whereby this parameter is maintained, that is, transporters, metabolism, etc.

2. The review is fully relevant to the field. The authors are experts
in the field, so their competence is out of doubt.  Although I find
the review largely tutorial, the text contains not only well
established concepts but also new and discussable arguments such as
the section 2, lines 80-104. There is no specific gap filled by this
study; it is a general overview.

3. The review summarizes the current knowledge in regulation of ionic balance in various situations, largely in cancer. One advantage of
the work is the methodological section 4. Evolution of the methods
for intracellular pH measurements in which the authors present an
overview of techniques of pH measurement. However, this section
should better fit the general flow of the text and be in line with
the title. This is why I suggested either to omit ‘cancer’ in the
title (which would broaden the horizon of the study) or re-model the
text flow to properly incorporate section 4. Maybe to make a brief
introduction and explain why the cancer-unrelated models are
discussed in this section.

4. Conclusions are clear-cut. I suggest making no separate section 3.5 and move its content to 6. Conclusion.

5. The references are sufficient for presenting the text as a comprehensive and timely review.

Reviewer 2 Report

Comments and Suggestions for Authors

The review ' On the importance of acidity in cancer cells and therapy' by Tafech et al has described the details about the effect of acidity on the cancer cells and its various mechanisms involved in tumor resistance and survival. It also put the points on the different pH regulators and how to measure the intracellular pH in tumor cells. This will surely help to understand the signaling and energetics of tumor cells for the treatment regime.

Minor revisions:

Line 53: Please explain how hypoxia is harmful for normal cells.

Line 146: What will happen to the tumor morphology (individual tumor cell and the 3D structure) if MCT4 is downregulated?

Line 510: Please change 'bye' to 'by'.

Line 575: Please provide the full form for 'ABC'.

Line 630: Please comment on the extracellular Ca+2 concentration.

Line 696: Please check the spelling 'inhibition'. Its written 'inibition'.

Comments on the Quality of English Language

It needs minor English edits.
